# HO-1089 and HO-1197, Novel Herbal Formulas, Have Antitumor Effects via Suppression of PLK1 (Polo-like Kinase 1) Expression in Hepatocellular Carcinoma

**DOI:** 10.3390/cancers15030851

**Published:** 2023-01-30

**Authors:** Yeonhwa Song, Su-Yeon Lee, Sanghwa Kim, Inhee Choi, Namjeong Kim, Jongmin Park, Haeng Ran Seo

**Affiliations:** 1Advanced Biomedical Research Lab, Institut Pasteur Korea, 16, Daewangpangyo-ro 712 beon-gil, Bundang-gu, Seongnam-si 13488, Gyeonggi-do, Republic of Korea; 2Medicinal Chemistry, Institut Pasteur Korea, 16, Daewangpangyo-ro 712 beon-gil, Bundang-gu, Seongnam-si 13488, Gyeonggi-do, Republic of Korea; 3H&O Biosis Co., Ltd., 19-10, Jeongnamsandan-ro, Jeongnam-myeon, Hwaseong-si 18514, Gyeonggi-do, Republic of Korea

**Keywords:** hepatocellular carcinoma (HCC), HO-1089, HO-1197, DNA damage, PLK1, cell cycle arrest

## Abstract

**Simple Summary:**

HO-1089 and HO-1197 have potential to be developed as herbal medicine or as part of combination therapy in HCC. Additionally, development of PLK1 inhibition as a therapeutic strategy for HCC will require more study into the selection of patients based upon molecular vulnerabilities and the development of mechanism-based, rationally-selected combination of herbs.

**Abstract:**

The treatment for hepatocellular carcinoma (HCC), a severe cancer with a very high mortality rate, begins with the surgical resection of the primary tumor. For metastasis or for tumors that cannot be resected, sorafenib, a multi-tyrosine protein kinase inhibitor, is usually the drug of choice. However, typically, neither resection nor sorafenib provides a cure. The drug discovery strategy for HCC therapy is shifting from monotherapies to combination regimens that combine an immuno-oncology agent with an angiogenesis inhibitor. Herbal formulas can be included in the combinations used for this personalized medicine approach. In this study, we evaluated the HCC anticancer efficacy of the new herbal formula, HO-1089. Treatment with HO-1089 inhibited HCC tumor growth by inducing DNA damage-mediated apoptosis and by arresting HCC cell replication during the G2/M phase. HO-1089 also attenuated the migratory capacity of HCC cells via the inhibition of the expression of EMT-related proteins. Biological pathways involved in metabolism and the mitotic cell cycle were suppressed in HO-1089-treated HCC cells. HO-1089 attenuated expression of the G2/M phase regulatory protein, PLK1 (polo-like kinase 1), in HCC cells. HCC xenograft mouse models revealed that the daily oral administration of HO-1089 retarded tumor growth without systemic toxicity *in vivo*. The use of HO-1197, a novel herbal formula derived from HO-1089, resulted in statistically significant improved anticancer efficacy relative to HO-1089 in HCC. These results suggest that HO-1089 is a safe and potent integrated natural medicine for HCC therapy.

## 1. Background

Hepatocellular carcinoma (HCC) is one of the most frequently occurring types of cancer. HCC effects are especially severe because HCC has a high mortality rate [1]. Among countries worldwide, the highest values for liver cancer incidence are in Eastern Asia and sub-Saharan Africa [2,3]. Currently, only angiogenesis inhibitors (five of which have been approved by the US FDA) and immune checkpoint inhibitors (two of which have been approved by the US FDA) are the two drug classes used for the treatment of HCC [4]. Neither approach is entirely effective against HCC.

One study found that compared to sorafenib, the use of bevacizumab with atezolizumab, a first-in-class combination, increased the overall survival and prolonged disease-free survival in patients with HCC [5]. Combining an immuno-oncology agent with an angiogenesis inhibitor changes cancer treatment from the use of an initial single medication to the use of combination therapy [6,7].

The use of herbal medicine could be explained as similar to combination cancer therapy because it involves multiple targets and multiple signaling pathways. The use of herbal medicine is gaining attention as a new strategy for cancer drug discovery [8] because it synergistically enhances the antitumor efficacy of various chemotherapeutic agents [9] while having fewer adverse effects [10]. Antitumor pathways activated by herbal medicine include blocking cell proliferation, the promotion of apoptosis, synergistic potentiation, reversal of resistance to chemotherapeutic or targeted drugs, reduction in side effects of Western medicine treatments, and improvements in immunity [11]. However, in most cases, the chemical and pharmacological mechanisms of herbal medicines are ambiguous. Most studies on the molecular mechanisms of anticancer activities of oriental medicines have been performed using active monomers or the crude extracts of single herbs.

In this study, we examined whether HO-1089 had a novel and effective therapeutic function when used for the treatment of HCC. HO-1089 is a novel herbal formulation that mainly consists of *Hedyotidis herba, Akebiae fructus*, prunella spike, *Curcuma zedoaria* Rosc. and Curcuma root. *Hedyotidis herba* can suppress the activation of the AKT/mTOR pathway in HCC [12] cells and during breast cancer treatment [13]. *Akebia trifoliate* has anticancer potency in HCC cells, partly via the induction of endoplasmic reticulum stress; it is also safe and effective for the treatment of fatty liver disease [14,15]. The spike of the prunella plant targets the Notch1, Notch2, and Bcl-2 proteins in carcinoma cells in the human colon to inhibit their proliferation. In HCC, prunella spike inhibits the PI3K/Akt/mTOR pathway [16,17]. *Curcuma zedoaria* Rosc. is effective in inhibiting melanoma cells and lung cancer cells [18]. The root of the plant inhibits the proliferation of HCC cells because it targets the expression of VEGF and the regulation of the miR-21/TIMP3 axis [19,20].

We manufactured HO-1089 using the 12 herbs. We then examined the anticancer effects of HO-1089 for the induction of HCC cell death and the inhibition of tumor growth *in vivo* using HCC xenograft mice models. We also examined the mechanism of action associated with the chemotherapeutic activity of HO-1089. Using the recombination of the composition of HO-1089, we also generated a comparatively more efficient novel herbal formula HO-1197.

## 2. Materials and Methods

### 2.1. Preparation of HO-1089 and HO-1197

The experiments were performed using HO-1089 and HO-1197. HO-1089 consists of *H. herba*, *A. fructus*, *C. zedoaria* Rosc., Curcuma root and prunella spike and a variety of other medicinal herbs purchased from Yeongcheon Herbal Market (Yeongcheon, Republic of Korea), as received from H&O Biosis (Hwaseong-si, Republic of Korea). HO-1197 consists of *H. herba*, *A. fructus*, *C. zedoaria* Rosc., Curcuma root, and prunella spike. A total of 150 g HO-1089 and HO-1197 was soaked in 1 L distilled water, boiled for 2 h using a 2-L flask (Saehanlab Co., Seoul, Republic of Korea), and then filtered through a polyester filter (8 µm, Hwa Shin Textile Filter Co., Seoul, Republic of Korea). After the first filtration, the solution was passed through a 0.5 µm polypropylene filter cartridge (Teail Industries Co., Gyeonggi-do, Republic of Korea). The HO-1089 and HO-1197 filtrate was freeze-dried and stored in a desiccator at 4 °C. The freeze-dried powder was dissolved in warm water for use in the in vitro experiments.

### 2.2. Cell Lines and Culture Conditions

Our experiments used the HCC cell lines Huh7, Hep3B, SNU449, and SNU475, which came from the Korean Cell Line Bank (Seoul, Republic of Korea). Cell lines Huh7, SNU449, and SNU475 were cultivated in Roswell Park Memorial Institute 1640 medium (RPMI 16400; Welgene, Daegu, Republic of Korea), and the Hep3B cells were cultivated in Minimum Essential Medium (MEM; Welgene, Daegu, Republic of Korea). We supplemented both cell media with heat-inactivated 10% fetal bovine serum (Gibco, Grand Island, NY, USA) and 1× penicillin/streptomycin antibiotics (Gibco). All cells were maintained at 37 °C in humidified atmosphere of 5% CO_2_.

### 2.3. Cell Cycle Analysis

We treated the cells with HO-1089 (5 mg/mL) for 0, 6, 12, or 24 h, washed the cells with Dulbecco’s phosphate-buffered saline (DPBS; Welgene), resuspended the pellets in 100 µL DBPS, and saved the supernatants. For analysis by flow cytometry (BD Biosciences, Franklin Lakes, NJ, USA), we treated the resuspended cells with 5 µL propidium iodide (Sigma-Aldrich, St. Louis, MO, USA) for 10 min and then washed them with DPBS.

### 2.4. Dose–Response Curves

We seeded 2 × 10^3^ Huh7, Hep3B, or SNU475 cells in each well of 384-well plates (Greiner Bio-One, Monroe, NC, USA). We incubated various combinations of fractions and HO-1089 (H1–H13) for 48 h. The total concentration of HO-1089 and the cell fractions was 50 mg/mL (2-fold dilution, 20-points) in water (*v/v*). After treating cells with HO-1089 for 48 h, we fixed them at room temperature for 10 min in 4% paraformaldehyde (Sigma-Aldrich), washed them twice with DPBS, and stained nuclei using Hoechst 33342 (Invitrogen, Eugene, OR, USA) for 10 min at R.T. with 20 µg/ml. For every well, we analyzed at least 1000 cells over five microscopic fields, starting at the center. Image analysis was performed with Operetta High Content Screening (HCS) system (Perkin Elmer, Waltham, MA, USA) and Harmony software (Perkin Elmer). We determined the cell counts for the experimental samples and normalized them to the control cells for the relative survival.

### 2.5. Protein Separation and Immunoblot Analysis

To extract the protein, we treated the cells with lysis buffer (Thermo Fisher Scientific, Waltham, MA, USA) on ice for 30 min and centrifuged them for 10 min at 13,200 rpm at 4 °C. We transferred the supernatants to microcentrifuge tubes (Corning, Corning, NY, USA) and added 5× sample buffer (BioSolutions, Seoul, Republic of Korea). We boiled the samples for 10 min, loaded equal protein (10–30 µg) in each well, separated the proteins using 8% or 10% (depending on protein size) SDS–polyacrylamide electrophoresis gels, transferred the protein bands onto a nitrocellulose membrane (Pall, Port Washington, NY, USA), and blocked the membranes with 5% skim milk (BD Bioscience) for 30 min at room temperature. We incubated the membranes with appropriate antibodies for 16 h at 4 °C. These included rabbit anti-cleaved caspase-3 (Abcam, Cambridge, UK), rabbit anti-human PARP (Cell Signaling Technology, Danvers, MA, USA), rabbit anti-human HO-1 (Cell Signaling Technology), rabbit anti-human NQO-1 (Cell Signaling Technology), mouse anti-human γ-H2AX (Merck Millipore, Darmstadt, Germany), rabbit anti-human SNAIL (Cell Signaling Technology), rabbit anti-human α-smooth muscle actin (Abcam), mouse anti-human CD133/1 (AC133, Miltenyi Biotec, Bergisch Gladbach, Germany), rabbit anti-human phospho-histone H3 (Ser10) (Cell Signaling Technology), rabbit anti-human UBE2C (Cell Signaling Technology), rabbit anti-human PLK1 (Cell Signaling Technology), rabbit anti-human CDC20 (Cell Signaling Technology), goat anti-human apolipoprotein AI (Abcam), mouse anti-human apolipoprotein B (Novus Biologicals, Minneapolis, MN, USA), or mouse anti-human β-actin (Sigma-Aldrich). We incubated the washed blots for 1 h with anti-mouse IgG, horseradish peroxidase-conjugated secondary antibody (Cell Signaling Technology) or anti-rabbit IgG, and horseradish peroxidase-conjugated secondary antibody (Cell Signaling Technology). We visualized the protein bands by enhanced chemiluminescence (Thermo Fisher Scientific) and recorded the chemiluminescent images on X-Omat AR films (Eastman Kodak Co., Rochester, NY, USA). β-actin was the control for each sample.

### 2.6. Sorafenib Combination Assay

We seeded 2 × 10^3^ Huh7 cells in each well of 384-well plates and added HO-1089 at 0, 1, or 5 mg/mL and sorafenib (Santa Cruz Biotechnology, Dallas, TX, USA) at a final concentration of 0, 1, or 3 µM for 48 h. Cells were fixed, stained with Hoechst 33342, and analyzed as described above for the dose–response curve experiments.

### 2.7. Reactive Oxygen Species Detection

We seeded 2 × 10^3^ Huh7 cells in each well of 384-well plates and added HO-1089 at 0, 1, 5, or 10 mg/mL for 24 h. We washed the cells with DPBS and detected reactive oxygen species (ROS) by incubating the cells with CM-H_2_DCFDA (Thermo Fisher Scientific) for 30 min. We used automated live-cell multispectral image acquisition to capture images with the HCS system using the CM-H_2_DCFDA optimum excitation wavelength of 485 ± 20 nm and emission wavelength of 515 ± 10 nm. After image acquisition, Aleaxa 488 intensity and counting the number of nuclei, the ROS value was obtained by calculating the average intensity per condition.

### 2.8. Cell Migration Assay

Huh7 and SNU475 cells were seeded into 6-well plates (Corning, NY, USA) at a density of 1 × 10^6^ cells/well to allow full confluency. After confluence was achieved, the monolayer of the cells was scratched with a yellow pipette tip, and the cells migrated into the gap during the subsequent 24 h period while being exposed to 0, 1, or 5 mg/mL HO-1089. Images were captured using a light microscope (Zeiss, Jena, Germany). The images of the migration were analyzed using Image J, which calculated an area at 0 h and 24 h of scratched area with no cells. The value was calculated with the formulation: wound closure (%) = (initial area − final area)/initial area × 100, initial = 0 h, final = 24 h.

### 2.9. Microarray Analysis

We used Affymetrix GeneChip^®^ Human Gene 2.0 ST Arrays to measure the gene expression with total Huh7 RNA isolated using the RNeasy Mini kit (Qiagen, Hilden, Germany) from cells that had been treated with 5 mg/mL HO-1089. We determined RNA quantity using a Nanodrop-1000 Spectrophotometer (Thermo Fisher Scientific) and RNA quality with the RNA 6000 Nano Chip and an Agilent 2100 Bioanalyzer (Agilent Technologies). We followed the Affymetrix protocol, converting 300 μg of RNA into double-stranded cDNA using a random hexamer that included a T7 promoter. We used T7 polymerase-mediated in vitro transcription (IVT) of the cDNA template to efficiently produce cRNA, which we purified using the Affymetrix sample cleanup module. We synthesized cDNA by reverse transcription with random primers using a dUTP-containing dNTP mix. We used uracil-DNA glycosylase and apurinic/apyrimidinic endonuclease restriction enzymes to fragment the cDNA and a terminal transferase reaction incorporating a biotinylated dideoxynucleotide to end label the DNA. We hybridized the short, end-labeled cDNA to the GeneChip^®^ Human Gene 2.0 ST array at 45 °C for 17 h, rotating at 60 rpm, as described in the Affymetrix Gene Chip Whole Transcript Sense Target Labeling Assay Manual. We stained and washed the chips using an Affymetrix GeneChip Fluidics Station 450 and scanned the chips with an Affymetrix GeneChip Array 3000 7G scanner. We extracted the data on the amount of gene expression from the scanned images using Affymetrix Command Console software, version 1.1, and stored the information as CEL files.

### 2.10. In Vivo Model of a Tumor Xenograft

Huh7 cells were injected into the abdominal region of six-week-old male Balb/c nude mice (*n* = 7) (2 × 10^6^ cells/mouse; subcutaneous injection). On day 7 after tumor inoculation, when tumors reached a volume of approximately 100 mm^3^, the mice were randomly divided into five groups (*n* = 7/group) and were administered saline (control), 100 mpk, 200 mpk, 500 mpk, or 800 mpk, respectively. HO-1089 and HO-1197 were dissolved in water and were administered daily for 3 weeks, via the oral route. Each mouse was observed for gross appearance and behavior, and body weights and tumor volumes were measured. Tumor volume was calculated using the formula: (L × I^2^)/2, where L = tumor length and I = tumor width. Dimensions were determined using calipers.

### 2.11. Statistical Analysis

We performed the experiments at least three times and present here the data as the mean ± the standard deviation. We used a Student’s *t*-test in Microsoft Excel to determine the statistical significance with the following values: * *p* < 0.05, ** *p* < 0.005, or *** *p* < 0.001.

## 3. Results

### 3.1. HO-1089 Induces Apoptosis in HCC

The five key herbs (H. herba, A. fructus, prunella spike, C. zedoaria Rosc., and Curcuma root) and a variety of other medicinal herbs that composed HO-1089 were obtained from a local suppliers. An expert certified herbalist examined the plants to verify that we used the correct plant species and parts. When we treated human HCC cell lines Huh7, Hep3B, and SNU475, we found dose-dependent HO-1089 cytotoxicity (Figure 1A). HO-1089 at 1 mg/mL and 5 mg/mL reduced cell viability by 21% and 40.2%, respectively. In contrast, we found that 3 µM sorafenib had little effect on cell viability. The analysis of cell viability found that, in HCC cells, the combination treatment of HO-1089 and sorafenib could be used as a sensitizer for the highly efficient treatment of sorafenib in a dose-dependent manner (Figure 1B).

To determine whether HO-1089-induced the inhibition of cell growth was associated with an increase in apoptosis, we used a sub-G1 assay to measure apoptosis. This assay is widely used as it is easy, rapid, reliable, reproducible, and economical. Using flow cytometry, we estimated the sub-G1 peak in HO-1089-treated Huh7 and SNU475 cells. We found that treatment with 5 mg/mL HO-1089 induced an increase in sub-G1 populations in both Huh7 and SNU475 cells (Figure 1C).

To investigate the specific apoptotic mechanisms induced by HO-1089, we evaluated the expression of cleaved caspase-3 and cleaved-poly (ADP-ribose) polymerase (PARP) in Huh7cells, Hep3B cells, SNU449 cells, and SNU475 cells using Western blot analysis. Caspase-3, a member of the interleukin-1 beta-converting enzyme family, is implicated in the induction of apoptosis. Therefore, cleaved caspase-3, which is an active form of caspase-3, is considered to be an ideal marker for the measurement of apoptosis. Levels of cleaved caspase-3 and cleaved PARP were increased in HO-1089-treated Huh7cells, Hep3B cells, SNU449 cells, and SNU475 cells (Figure 1D).

### 3.2. HO-1089 Induces DNA Damage by ROS Accumulation in HCC

Because ROS can provide important clues about the physiological status of the cell, we assessed its production to determine the mechanism of HO-1089-induced cytotoxicity in Huh7 cells and SNU475 cells. HO-1089 induced a significantly high production of ROS in a dose-dependent manner. We detected this change using the permeable and redox-sensitive dye, CM-H_2_DCFDA, after indicated treatment times with HO-1089 in Huh7 cells and SNU475 cells (Figure 2A,B). However, HO-1089 did not enhance the expression of the antioxidant heme oxygenase 1 (HO-1), or of NAD(P)H quinone oxidoreductase 1 (NQO-1), at a wide range of concentrations in Huh7 cells and SNU475 cells (Figure 2C).

Because ROS are well recognized as mediators of DNA damage, we examined the expression levels of γ-H2AX after treatment with HO-1089 in HCC cells. γ-H2AX is a sensitive molecular marker of DNA damage and repair. As the amount of ROS increased due to HO-1089 treatment, the expression of γ-H2AX also increased in Huh7 cells and SNU475 cells (Figure 2D).

### 3.3. HO-1089 Attenuates Migration of HCC Cells

We also estimated the metastatic capacity in Huh7 cells and SNU475 cells after HO-1089 treatment using a cell migration assay. We found that HO-1089 suppressed the metastatic capacity in Huh7 cells and SNU475 cells in a dose-dependent manner (Figure 3A). HO-1089 treatment resulted in an increasing expression of EMT-related molecules such as Snail1 in Huh7 cells and SNU475 cells, and α-smooth muscle actin in Huh7 (Figure 3B).

In a previous study, we characterized the CD133^+^ cells in primary HCC cells [21]. Because CD133 cells overexpress and have increased migration ability in HCC cells, we estimated the CD133 expression using Western blot analysis after treatment with HO-1089 in HCC cells. HO-1089 treatment did not affect the expression of CD133 (Figure 3B).

### 3.4. HO-1089 Induces Cell Cycle Arrest via Suppression of Mitosis-Related Proteins in HCC

Determining the mechanisms of drug action in human cells remains a major challenge. To identify the potential targets of HO-1089, we used microarray analysis to examine genome-wide alterations in gene expression in HO-1089-treated HCC cells.

Biological pathways in fold quantity-based analyses are more detailed. This study revealed that the HO-1089 treatment resulted in pathway up-regulation and down-regulation (Appendix A). Biological pathways involved in metabolism were significantly suppressed in HO-1089-treated HCC cells; significant increases in biological pathways using HO-1089 treatment were not found (Figure 4A). As a result of the analysis of the biological pathways down-regulated by HO-1089 treatment, metabolism and mitotic cell cycle-related biological pathways were markedly suppressed (Appendix A). Treatment with HO-1089 resulted in defects in biological pathways involved in DNA damage regulation and overall cell cycle regulation, especially in the mitotic cell cycle (Figure 4B). Using expression levels ≥3 as arbitrary cutoffs for authentic gene expression, we found 20 genes that had decreases in expression after HO-1089 treatment (Table 1). The results for mitotic spindle checkpoint and mitotic nuclear division-related genes, such as PLK1 (polo-like kinase 1), ubiquitin-conjugating enzyme E2C (UBE2C), and cell division cycle 20 homolog, and lipid metabolism-related genes, such as apolipoprotein A2 (APOA2), apolipoprotein B (APOB), and apolipoprotein H (APOH), are presented in Table 1.

We found alterations in the protein expressions of down-regulated genes after HO-1089 treatment. Treatment with HO-1089 sufficiently inhibited PLK1 and UBE2C expression, whereas APOA and APOB expression were not changed after HO-1089 treatment in Huh7 cells, Hep3B cells, or SNU475 cells (Figure 4C). PLK1 and UBE2C participate in a broad range of processes from the start to the completion of mitosis. Along with the inhibition of PLK1 and UBE2C expression, the phosphorylation of histone H3 on Ser-10 (an epigenetic mitotic marker) was also inhibited by HO-1089 treatment in HCC cells (Figure 4C). To elucidate the mechanisms of HO-1089-mediated increased apoptosis, we examined the cell cycle distribution after HO-1089 treatment. Cell cycle analysis revealed 36.7% G2/M arrest in control cells and 47.3% G2/M arrest in HO-1089-treated HCC cells (Figure 4D).

PLK1 overexpression occurs in a variety of human cancers; it is also associated with a poor prognosis. Therefore, the dose–response curves of HO-1089 were used to determine the effective concentrations required to decrease proliferation by 50% (EC_50_) in a variety of human cancer cell lines. HO-1089 had an overall even anticancer effect in a variety of human cancer cell lines (Table 2).

### 3.5. HO-1197, a Novel Herbal Formula, Has a Better Improved Anticancer Efficacy Than HO-1089

To analyze the capability of HO-1089 to suppress HCC *in vivo*, we transplanted Huh7 cells into BALB/c nude mice. The oral administration of HO-1089 significantly reduced the tumor volume without a loss in body weight (Figure 5A,B). We found no significant differences in the levels of AST or ALT between HO-1089-treated mice, sorafenib-treated mice, or saline-treated mice (Figure 5C).

We generated 12 new herbal formulas. Each was made by combining the five key ingredients of HO-1089 (*H. herba*, *A. fructus*, prunella spike, *C. zedoaria* Rosc., and Curcuma root) to reduce the total administered volume of the drug and to enhance the anticancer efficacy. Using the dose–response analyses of the 12 herbal formulas, we selected HO-1197 as the best herbal combination for HCC therapy (Figure 6A).

Treatment with HO-1197 also sufficiently attenuated the expression of PLK1 and cdc20 and induced apoptosis in HCC in Huh7 cells and SNU475 cells. HO-1197 had a similar mode of action as HO-1089 (Figure 6B). The oral administration of HO-1197 also resulted in statistically significant tumor regression in Huh7 cell-transplanted BALB/c nude mice without alteration in the levels of AST or ALT (Figure 6C).

## 4. Discussion

Herbs are increasingly recognized as valuable treatments for cancers. This is reflected by the traditional use of herbal medicines to treat liver cancer. The use of herbal medicines can effectively improve the quality of life in patients with cancer-associated cachexia [22]. For these reasons, patients with cancer undergo at least one type of complementary or alternative therapy after a cancer diagnosis [23,24]. In many cases, because there is no reliable scientific evidence that use of herbal remedies alone can cure or treat cancer, they are used in combination with conventional treatments for cancer therapy [25,26].

In this study, we evaluated the anticancer activity of the herbal formula HO-1089 without combining it with conventional treatment for HCC. We found that HO-1089 had a high anticancer activity used alone in vitro and *in vivo*.

Most herbal medicines have anticancer properties with ambiguous pharmacological mechanisms. Therefore, the identification of the molecular mechanisms associated with the anticancer action of these formulations is a major challenge for liver cancer treatment. ROS alters the mitochondrial metabolic activity. They also affect apoptotic pathways via the accumulation of DNA damage. High levels of accumulation of cellular ROS using treatment with HO-1089 significantly induced increasing DNA damage and attenuated the migration ability in HCC.

To determine the global transcriptional effects of HO-1089 in HCC, we used an RNA-sequencing approach that significantly accelerated the process of drug target identification. RNA-seq revealed that HO-1089 had a clear anticancer mechanism. It controlled the mitotic cell cycle during the complex process of HCC growth signaling. PLK1 and UBE2C were HO-1089 targets and were essential for the induction of cell cycle arrest and cell death in HCC cells.

The cell cycle protein PLK1, which promotes mitosis [27,28], is often overexpressed in cancer and may predict small-cell lung cancer, colon cancer, ovarian cancer, and HCC [29,30]. The overexpression of PLK1 may override mitotic checkpoints, thereby dysregulating oncogenic cell proliferation [31]. The inhibition of PLK1 in cancers has been shown to result in the arrest of mitosis, damage to DNA, and static tumors [32]. Some small-molecule inhibitors of PLK1 were shown to result in the arrest of mitosis with subsequent cell death [33]. Similarly, we demonstrated here that the HO-1089 treatment of HCC cells inhibited PLK1 and resulted in DNA damage and mitotic arrest.

UBE2C exerts oncogenic effects on various human solid cancers such as HCC [34], breast cancer [35], nasopharyngeal carcinoma [36], and colon cancer [37]. Like PLK1, UBE2C is also essential for mitotic cell cycle regulation to accelerate the cell proliferation and malignant transformation [38]. We found that HO-1089 inhibited PLK1 and UBE2C. This result suggested that HO-1089 is a potential therapy for HCC and other types of cancers. For the development of herbal medicine-based anticancer drugs, we tried to find the best herbal medicine combinations and ratios based on the core ingredients of HO-1089, and we will continue to do so.

## 5. Conclusions

These results suggested a potential therapeutic value for HO-1197, a novel herbal formula derived from HO-1089, as an herbal medicine or as part of a combination therapy. However, the future success of PLK1 inhibitors as an HCC therapy will require additional information on the particular molecular vulnerability of patients and an understanding of the mechanisms of the disease in order to drive a rational approach for selecting the appropriate combinations of herbs.

## Figures and Tables

**Figure 1 cancers-15-00851-f001:**
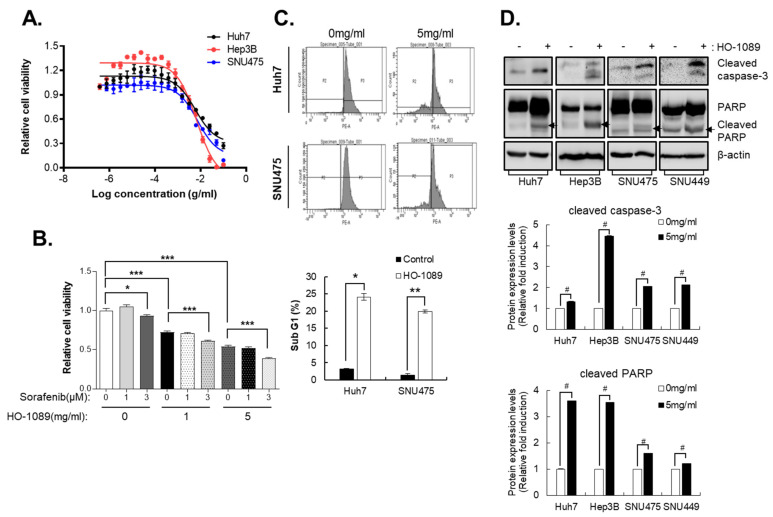
Apoptosis induced by HO-1089 treatment. (**A**) Huh7, Hep3B, and SNU475 cells were treated with HO-1089 for 48 h beginning with 50 mg/mL and using sequential 2-fold serial dilutions. Cell viability was measured by the number of cell nuclei. (**B**) The viability of Huh7 cells treated for 48 h with 0, 1, or 3 µM sorafenib or 0, 1, or 5 mg/mL HO-1089 alone or together. (**C**) The percentage of sub-G1 populations in Huh7 or SNU475 cells treated for 48 h with 0 or 5 mg/mL HO-1089 was determined by flow cytometry. (**D**) Cleaved caspase-3 and cleaved PARP, which indicate apoptosis, were identified by immunoblots (upper panel) in Huh7, Hep3B, SNU449, and SNU475 cells treated for 48 h with 5 mg/mL HO-1089. The Western blot images were quantitatively analyzed (lower panel). Data are expressed as the means ± SD (*n* = 3). * *p* < 0.05, ** *p* < 0.005, # *p* < 0.0005, *** *p* < 0.0001 compared to the control group.

**Figure 2 cancers-15-00851-f002:**
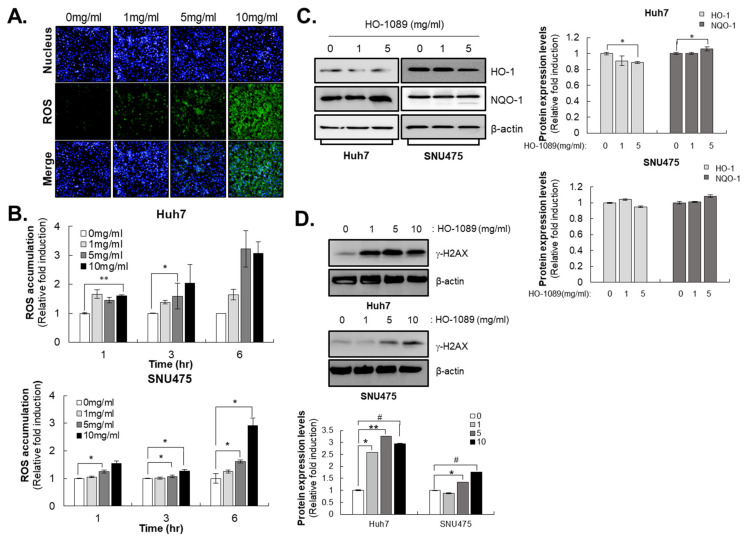
DNA damage by ROS accumulation induced by HO-1089 treatment: (**A**,**B**) Huh7 cells were treated with HO-1089 concentrations of 0, 1, 5, or 10 mg/mL for 24 h. ROS were detected with CM-H_2_DCFDA staining and analyzed using an HCS system. ROS intensity was measured and calculated. (**C**) The expression of antioxidant-related proteins was examined in 0, 1, or 5 mg/mL HO-1089-treated Huh7 and SNU475 cells (left panel). The Western blot images were quantitatively analyzed (right panel). (**D**) DNA damage-related protein, r-H2AX, was detected in 0, 1, 5, or 10 mg/mL HO-1089-treated Huh7 and SNU475 cells (upper panel). The Western blot images were quantitatively analyzed (lower panel). Results were normalized to control values to obtain values for relative fold induction. Data are expressed as means ± SD (*n* = 3). * *p* < 0.05, ** *p* < 0.005, # *p* < 0.0005 compared to the control group.

**Figure 3 cancers-15-00851-f003:**
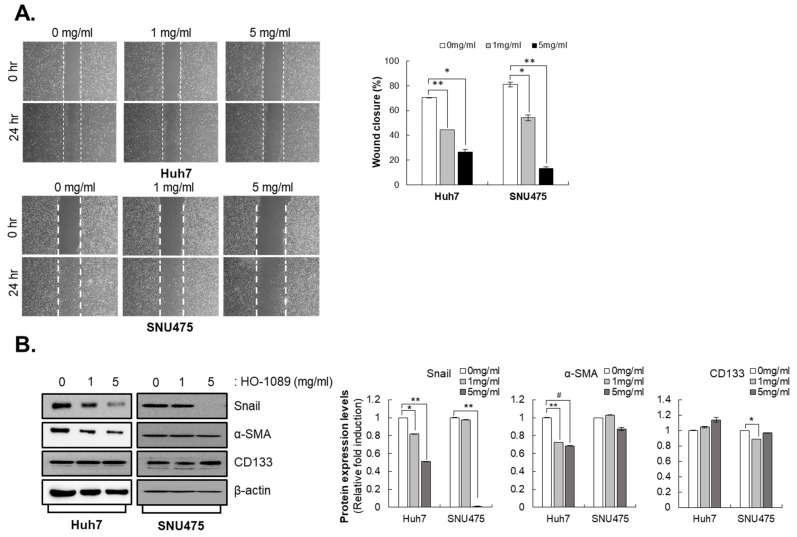
Inhibition of metastasis capacity by HO-1089 treatment: (**A**) Migration capacities of Huh7 and SNU475 cells were examined after treatment concentrations of HO-1089 at 0, 1, or 5 mg/mL for 24 h (left panel). The percentage of wound closure, which is calculated the area with no cells using image J software (right panel). (**B**) The EMT-related proteins, Snail and a-smooth muscle actin (SMA) and liver cancer stemness protein CD133, were detected when Huh7 and SNU475 cells were treated with 0, 1, or 5 mg/mL HO-1089 for 48 h (left panel). The Western blot images were quantitatively analyzed (right panel). Data are expressed as means ± SD (*n* = 3). * *p* < 0.05, ** *p* < 0.005, # *p* < 0.0005 compared to the control group.

**Figure 4 cancers-15-00851-f004:**
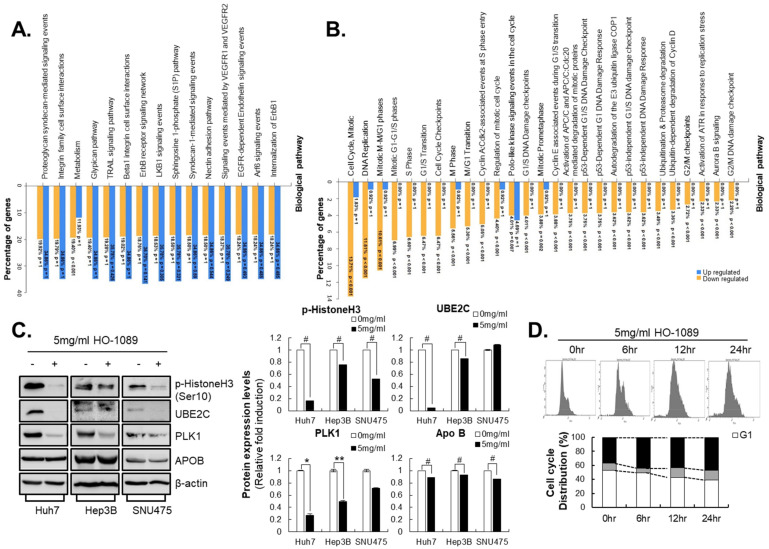
Suppression of mitosis-related protein induced by HO-1089 treatment: (**A**,**B**) The biological pathway of HO-1089 was analyzed using microarray analysis. Huh7 cells were treated with 5 mg/mL HO-1089 for 48 h. Cell cycle and mitosis-related gene expression were calculated and compared. The graphs show the percentages of genes that are not significantly changed (**A**), or significantly hanged (**B**), as well as their expression levels. (**C**) The genes from microarray analysis for which HO-1089 exposure changed the expression levels were examined in Huh7, Hep3B, and SNU475 cells (left panel). The Western blot images were quantitatively analyzed (right panel). (**D**) Cell cycle distributions induced by HO-1089 treatment in Huh7 cells were calculated at 0, 6, 12, and 24 h after treatment. Data are expressed as means ± SD (*n* = 3). * *p* < 0.05, ** *p* < 0.005, # *p* < 0.0005 compared to the control group.

**Figure 5 cancers-15-00851-f005:**
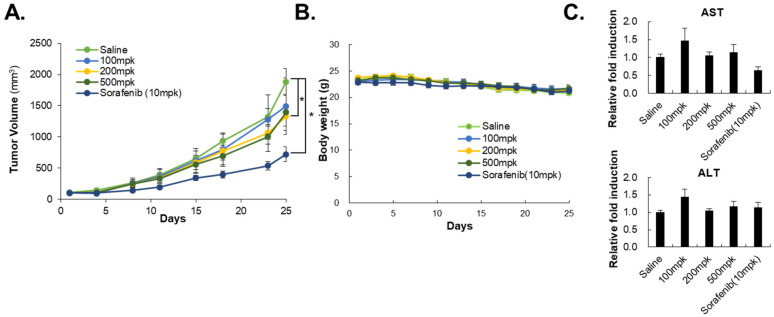
*In vivo* efficacy of HO-1089 treatment: (**A**) Tumor volume was calculated for 3 weeks, with HO-1089 treatment. Ten mpk sorafenib was used as a positive control. (**B**) Body weight was measured during the experimental period. (**C**) Liver toxicity present at the end of the *in vivo* experiment was examined using blood samples taken for the detection of AST and ALT levels. Relative fold induction was normalized to each control. Data are expressed as means ± SD (*n* = 7). * *p* < 0.05 compared to the control group.

**Figure 6 cancers-15-00851-f006:**
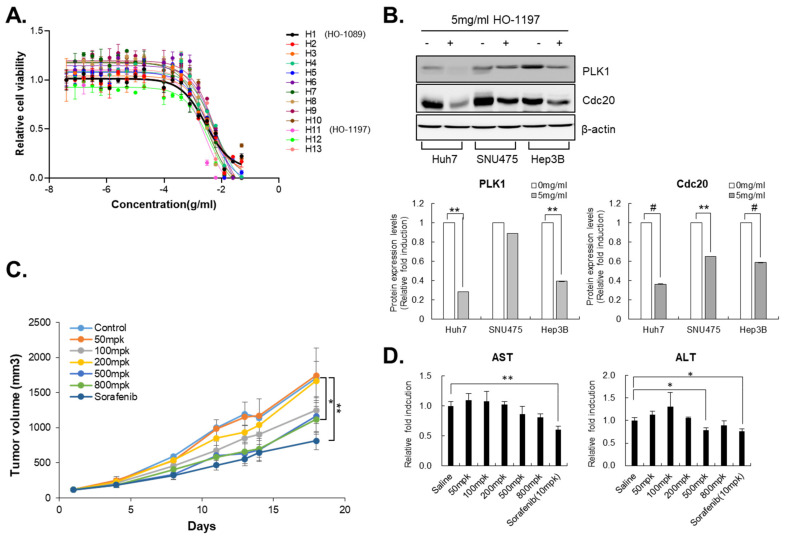
Key herbal combination, HO-1197, showed anticancer efficacy in HCC: (**A**) HO-1089 (H1) and 12 combinations of key ingredients including HO-1197 (H11) were used to treat HCC cells, from 50 mg/mL using 2-fold dilutions, for 48 h. Cell viability was measured using the counts of cell nuclei. (**B**) Expression levels of PLK1 and Cdc20 were examined in Huh7, Hep3B, and SNU475 cells after treatment with 5 mg/mL HO-1197 for 48 h (upper panel). The Western blot images were quantitatively analyzed (lower panel (**C**,**D**). *In vivo* efficacy of HO-1197 was examined using observation of changes in tumor volume (**C**) and liver toxicity (**D**) for 3 weeks. Data are expressed as means ± SD (*n* = 3). * *p* < 0.05, ** *p* < 0.005, # *p* < 0.0005 compared to the control group.

**Table 1 cancers-15-00851-t001:** Treated Huh7.

No	Log2Ratio1	Gene Name	Protein
1	−4.7104928	HIST1H2AI	histone cluster 1, H2ai
2	−4.235694	ID1	inhibitor of DNA binding 1, dominant negative helix-loop-helix protein
3	−4.0174427	KNG1	kininogen 1
4	−3.903392	ZNF93	zinc finger protein 93
5	−3.821343	ORM2	orosomucoid 2
6	−3.7330516	HIST1H2BM	histone cluster 1, H2bm
7	−3.505795	APOB	apolipoprotein B
8	−3.503174	F8A1	coagulation factor VIII-associated 1
9	−3.465386	HIST1H3I	histone cluster 1, H3i
10	−3.3550105	AGMAT	agmatinase
11	−3.288377	UBE2C	ubiquitin-conjugating enzyme E2C
12	−3.236855	IFITM2	interferon induced transmembrane protein 2
13	−3.230311	PLK1	polo-like kinase 1
14	−3.194956	NAR-E	small ILF3/NF90-associated RNA E
15	−3.133499	HABP2	hyaluronan binding protein 2
16	−3.1148475	NCOA5	nuclear receptor coactivator 5
17	−3.097753	APOA2	apolipoprotein A-II
18	−3.097341	CDCA8	cell division cycle associated 8
19	−3.094738	APOH	apolipoprotein H
20	−3.073547	CDC20	cell division cycle 20

**Table 2 cancers-15-00851-t002:** Value of HO-1089 on the growth of human cancer cell lines.

Name of Cell Line	IC_50_ (mg/mL)
Huh7	2.75
H460	8.92
A549	5.49
AGS	1.5
HT29	2.27
MDA-MB231	2.18
SNU213	14.5
A375	1.9

## Data Availability

Data available in a publicly accessible repository that does not issue DOIs. Publicly available datasets were analyzed in this study. This data can be found here: [link/accession number].

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
