# Peer review of "HO-1089 and HO-1197, Novel Herbal Formulas, Have Antitumor Effects via Suppression of PLK1 (Polo-like Kinase 1) Expression in Hepatocellular Carcinoma"

_cancers, 2023, doi:10.3390/cancers15030851_

Round 1

Reviewer 1 Report

In this article, Song et al. reported a new integrated natural medicine for HCC therapy. In particular, author evaluated the hepatocellular carcinoma (HCC) anticancer efficacy of the new herbal formula HO-1089. The author showed that the treatment with HO-1089 inhibited HCC tumor growth by inducing DNA-damage-mediated apoptosis, by arresting HCC cell replication during G2/M phase, and attenuated expression of PLK1. Moreover, HO-1089 attenuated the migratory of HCC cells via inhibition of expression of EMT-related proteins. The authors also analyzed HO-1089 effects in vivo models showing a retarded tumor growth without systemic toxicity. Finally, Song et al. showed that HO-1197, a formula derived HO-1089, improved anticancer efficacy relative to HO-1089 in HCC, suggesting that HO-1197 is a safe and potent integrated natural medicine for HCC therapy.

I recommend the reconsideration of this paper only after considering the following comments and major revisions:

1. In lines 224-226, the authors write that “Levels of cleaved caspase-3, and cleaved PARP were increased in HO-1089-treated Huh7cells, Hep3B cells, SNU 449cells and SNU475 cells (Fig. 1D)”. In my opinion these results are clear for cleaved caspase-3 but not for cleaved PARP. In particular, the PARP cleavage is evident in cell lines Huh7 and Hep3B but not in SNU 449 and SNU475 cell lines. In SNU475 cells there is not a relevant effect of HO-1089 on PARP cleavage. The authors should include the histogram with relative statistical analysis.

2. In lines 200-202, the authors write: “HO-1089 at 1 mg/mL and 5 mg/mL reduced cell viability by 21% and 40.2%, respectively. In contrast, we found that 3 μM sorafenib had little effect on cell viability [Figure 1-B].” Why the authors choose the concentrations of 1 and 3μM for Sorafenib treatment? Sorafenib is the drug used for treatment of hepatocellular carcinoma but what is the concentration used for clinical treatment?

3. In line 257-260, the authors write “In a previous study, we…”. The authors should insert the relative reference. Moreover, they assess that HO-1089 treatment did not affect expression of CD133 (Fig. 3-B). This result is clear in Huh7 cells but in western blot of SNU475 cells, CD133 seems up-regulated at 5mg/ml of HO-1089. The authors should justify this evidence.

4. In Fig. 4-C, the authors show the effects of HO-1089 on different protein in 3 cell lines. The western blot referred to APO A in Hep3B cell line in not clear. In particular the signal of protein is not well reveled. The authors should insert an image in which the signal is clearer.

5. In lines 308-312 and in Table 3, the authors write: “PLK1 overexpression occurs in a variety of human cancers; it is also associated with a poor prognosis. Therefore, dose-response curves of HO-1089 were used to determine the effective concentrations required to decrease proliferation by 50% (EC 50) in a variety of human cancer cell lines. HO-1089 had an overall even anticancer effect in a variety of human cancer cell lines [Table II].” Although it is clear that PLK1 overexpression occurs in a variety of human cancers, I don’t understand why the authors analyzed HO-1089 effects on proliferation in different human cancer cell lines (for example, breast cancer or colorectal adenocarcinoma) which are not objects of the present study and on which no other experiments were carried out. The authors should discuss this choice.

6. In my opinion Fig. 6-A is not clear. There are too many curves in the same image and it is not showed which is the curve related to HO-1197. I think that H1, so the black line, represent the effect of HO-1089 on cell survival, but I don’t understand which curve represent HO-1197 effect. The authors should modify the image and should insert another image in which show the curve related to HO-1089 and HO-1197 effects, in order to justify choice of HO-1197.

7. In lines 333-335 the authors said: “Treatment with HO-1197 also sufficiently attenuated expression of PLK1 and cdc20 and induced apoptosis in HCC in Huh7 cells and SNU475 cells. HO-1197 had a similar mode of action as HO-1089 [Figure 6-B].” The data reported in Fig. 6-B shown that HO-1197 induces a decrease of PLK1 levels both in Huh7 cells that Hep3B cells but not in SNU475 cells, in different manner respect to HO-1089. The authors should justify this data. Moreover, the authors write that HO-1197 induced apoptosis in similar mode of action of HO-1089 but no experiments are reported on apoptosis effects of HO-1197. The authors should show the effects of HO-1197 on cleavage of caspase 3, of PARP, and perform a sub-G1 assay to measure apoptosis.

8. Discussion paragraph is incomplete. The authors do not discuss the results obtained with HO-1197. All data need to be discussed.

9. In conclusion paragraph the authors write: “These results suggested a potential therapeutic value of HO-1197…”. I think that the experiments proposed in this article on HO-1197 effects are too few to say this. The authors should improve the experiments on HO-1197 effects.

Author Response

Thank you so much for taking the time to read the paper we submitted and for your comments and advice. In answering your comments and questions, I felt a lot of lack of paper. I hope the modified figures and manuscript will satisfy you.

Reviewer 2 Report

 Song et al. intended to evaluate the HCC anticancer efficacy of the new herbal formula, HO-1089 in their study titled "HO-1197, a novel herbal formula, has antitumor effects via suppression of PLK1 (polo-like kinase 1) expression in hepatocellular carcinoma." 

This constitutes a large, comprehensive, and broadly rational body of review work that is appreciable. However, there are several major concerns that can be resolved to improve the quality of the manuscript, listed below.

1.      In the abstract line No. 29 why is the treatment mentioned as HO-1197?

2.      Please mention the error bar and significance value in figure 1C.

3.      Why is the cell viability portrayed differently in Figures 1B & E? I would suggest putting it together as this is the separate and combined observation of sorafenib and HO-1089 treatment.

4.      Please put an improved clear representative picture of ROS staining, Figure 2A. Also please mention the quantitative procedure in the figure legend.

5.      All figure legends should be rewritten with clear information.

6.      Please put the statistical significance value in figure No 2B.

7.      In lines No 254-256 what the authors meant to state? There are so many places where the meaning is not clear. I would suggest the authors to revisit the manuscript carefully and rewrite the portions.

8.      Please mention the reference No in line No. 257.

9.      In figure No. 4 please remove the ‘FIGURE 4’ denotation from the figure. The text of Figures 4A and B should be from the left to right direction.

10.   The Xenograft tumor pictures should be given at least in the supplementary figure.   

11.   I strongly suggest to please improve the writing quality of the whole manuscript. (Example: lines No. 47, 70, 197, etc.).

Author Response

(The authors gave the same response as above.)

Reviewer 3 Report

This study investigated the potential antitumor effects of a novel herbal formula, HO-1089, which stimulated apoptosis and ROS accumulation, suppressed migration, and induced cell cycle arrest in HCC cell lines in vitro. In vivo model of a tumor xenograft furtherly demonstrated its antitumor activity. The study contains data of potential interest. There remain however issues to be resolved before proceeding further.

Major comments:

1)    In Figure 6A, Line 328-330, the detailed ingredients and concentration of the 12 new herbal formulas, and how HO-1197 was selected as the best herbal combination should be clearly described. As there are too many lines in Figure 6A, it’s hard to tell the difference. Through this manuscript, HO-1197 seems to be a simplification of HO-1089, while the antitumor activity of HO-1197 was better, please discuss the possible reason.

2)    Most importantly, the respective antitumor effect of the five key ingredients of HO-1089 are suggested to be determined separately. Understanding the function of each drug in the herbal formula will be helpful to promote its development and clinical application.

3)    As herbal formula is made of a lot of ingredients, the potential active compounds of this formula are suggested to be discussed.

4)    Most of this study were talking about the effects of HO-1089, so the title is suggested to change to HO-1089.

5)    Quantification for the immunoblot analysis are required.

6)    In Figure 2B and 5A, the statistical significance markers were missing.

7)    In Figure 1A, cell viability was measured by the number of cell nuclei, however, this method cannot separate the live cells from dead cells. MTT or CCK8 might be better for cell viability assay.

8)    In Figure 2C, NQO-1 is not described in the text.

9)    In Figure 2D, Line 240, what is γ-H2AX (a sensitive molecular marker of DNA damage) should be described. In addition, direct determination of DNA damage using a commercial kit would be better than only check the marker.

10) In Figure 3A, quantification for the scratch assay is required. 

Minor comments:

1)    The resolution of Figure 4A and 4B is low.

2)    Author contributions and Conflicts of Interest are missing.

3)    Line 89-96, the incubate condition (37 °C, 5% CO2) should be described.

Author Response

(The authors gave the same response as above.)

Round 2

Reviewer 1 Report

In this article, Song et al. reported a new integrated natural medicine for HCC therapy. In particular, author evaluated the hepatocellular carcinoma (HCC) anticancer efficacy of the new herbal formula HO-1089. The author showed that the treatment with HO-1089 inhibited HCC tumor growth by inducing DNA-damage-mediated apoptosis, by arresting HCC cell replication during G2/M phase, and attenuated expression of PLK1. Moreover, HO-1089 attenuated the migratory of HCC cells via inhibition of expression of EMT-related proteins. The authors also analyzed HO-1089 effects in vivo models showing a retarded tumor growth without systemic toxicity. Finally, Song et al. showed that HO-1197, a formula derived HO-1089, improved anticancer efficacy relative to HO-1089 in HCC, suggesting that HO-1197 is a safe and potent integrated natural medicine for HCC therapy.

I recommend the reconsideration of this paper only after considering the following comments and major revisions:

1.      In lines 277-279, the authors write: “HO-1089 treatment resulted in increasing expression of EMT-related molecules such as α-smooth muscle actin and Snail1 in Huh7 cells and SNU475 cells [Figure 3-B].” This sentence is in contrast with the histograms present in the Figure 3B, which the authors insert in this revised version of the paper. The figure 3B shows that Snail protein level decreases after treatment with HO-1089 in both cell lines, while no changes are present for the α-SMA protein level after treatment in both cell lines. The authors have to review these data and this part of the work.

2.      In lines 280-283, the authors write: “In a previous study, we characterized CD133+ cells in primary HCC cells [21]. Because CD133 cells overexpress and have increased migration ability in HCC cells, we estimated CD133 expression using western blot analysis after treatment with HO-1089 in HCC cells. HO-1089 treatment did not affect expression of CD133 [Figure 3-B].” This last sentence is in contrast with the histogram present in the Figure 3B, which the authors insert in this revised version of the paper. The data reported showed that HO-1089 treatment did not affect expression of CD133 in Huh 7 cell line, while HO-1089 treatment decrease CD133 protein level in SNU 475 cell line, in particular CD133 protein level decreases after treatment with HO-1089 at concentration of 1mg/ml but not at 5mg/ml. The authors have to review these data and this part of the work. Moreover, they must to justify the results obtained to CD133 protein in SNU 475 cell line.

3.      in the previous review I wrote: “In Fig. 4-C, the authors show the effects of HO-1089 on different protein in 3 cell lines. The western blot referred to APO A in Hep3B cell line in not clear. In particular the signal of protein is not well reveled. The authors should insert an image in which the signal is clearer.” The authors’ response is: “Apolipoprotein A has different patterns for each HCC cell line. Apolipoprotein A is not expressed in Hep3B. In this study, Apo A was deleted from this figure because there was no change in expression according to HO-1089 treatment.” In the Figure is still present APO western blot. The authors must change the figure and the data reported in the results section.

4.      In lines 365-367, the authors write: “Treatment with HO-1197 also sufficiently attenuated expression of PLK1 and cdc20 and induced apoptosis in HCC in Huh7 cells and SNU475 cells. HO-1197 had a similar mode of action as HO-1089 [Figure 6-B].” The data reported in figures 4C and 6B showed that, respectively, HO-1089 and HO-1197 attenuates expression of PLK1 in Huh7 and Hep3B cell lines but not in SNU475 cells, contrary to what is written by the authors. Moreover, the authors don’t show the effect of HO-1089 on cdc20 protein level, thus the two compounds have a similar effect only on PLK1 protein. The authors must insert the data on cdc20 protein after treatment with HO-1089 and modify what is written.

Reviewer 2 Report

The authors have addressed all the concerns which have been pointed out.

Hope their research will be followed in the future.

Author Response

Dear Reviewer,

Thank you so much for taking the time to review our manuscript and figures, and your comments will be of great help in future research. Thank you so much.

Sincerely,

Haeng Ran Seo

Reviewer 3 Report

Thank you for your careful and comprehensive response.

I recommend to accept in present form.

Author Response

(The authors gave the same response as above.)

Round 3

Reviewer 1 Report

Thank you for fulfilling my comments!